# On the Hardness of Robust Classification

**Pascale Gourdeau**
University of Oxford
pascale.gourdeau@cs.ox.ac.uk

**Varun Kanade**
University of Oxford
varunk@cs.ox.ac.uk

**Marta Kwiatkowska**
University of Oxford
marta.kwiatkowska@cs.ox.ac.uk

**James Worrell**
University of Oxford
james.worrell@cs.ox.ac.uk

## Abstract

It is becoming increasingly important to understand the vulnerability of machine learning models to adversarial attacks. In this paper we study the feasibility of robust learning from the perspective of computational learning theory, considering both sample and computational complexity. In particular, our definition of robust learnability requires polynomial sample complexity. We start with two negative results. We show that no non-trivial concept class can be robustly learned in the distribution-free setting against an adversary who can perturb just a single input bit. We show moreover that the class of monotone conjunctions cannot be robustly learned under the uniform distribution against an adversary who can perturb $\omega(\log n)$ input bits. However if the adversary is restricted to perturbing $O(\log n)$ bits, then the class of monotone conjunctions can be robustly learned with respect to a general class of distributions (that includes the uniform distribution). Finally, we provide a simple proof of the computational hardness of robust learning on the boolean hypercube. Unlike previous results of this nature, our result does not rely on another computational model (e.g. the statistical query model) nor on any hardness assumption other than the existence of a hard learning problem in the PAC framework.

## 1 Introduction

There has been considerable interest in adversarial machine learning since the seminal work of Szegedy et al. [25], who coined the term *adversarial example* to denote the result of applying a carefully chosen perturbation that causes a classification error to a previously correctly classified datum. Biggio et al. [4] independently observed this phenomenon. However, as pointed out by Biggio and Roli [3], adversarial machine learning has been considered much earlier in the context of spam filtering [8, 19, 20]. Their survey also distinguished two settings: *evasion attacks*, where an adversary modifies data at test time, and *poisoning attacks*, where the adversary modifies the training data.[1]

Several different definitions of adversarial learning exist in the literature and, unfortunately, in some instances the same terminology has been used to refer to different notions (for some discussion see e.g., [11, 10]). Our goal in this paper is to take the most widely-used definitions and consider their implications for robust learning from a statistical and computational viewpoint. For simplicity, we will focus on the setting where the input space is the boolean hypercube $\mathcal{X} = \{0,1\}^n$ and consider the *realizable* setting, i.e. the labels are consistent with a target concept in some concept class.

An *adversarial example* is constructed from a *natural example* by adding a perturbation. Typically, the power of the adversary is curtailed by specifying an upper bound on the perturbation under some

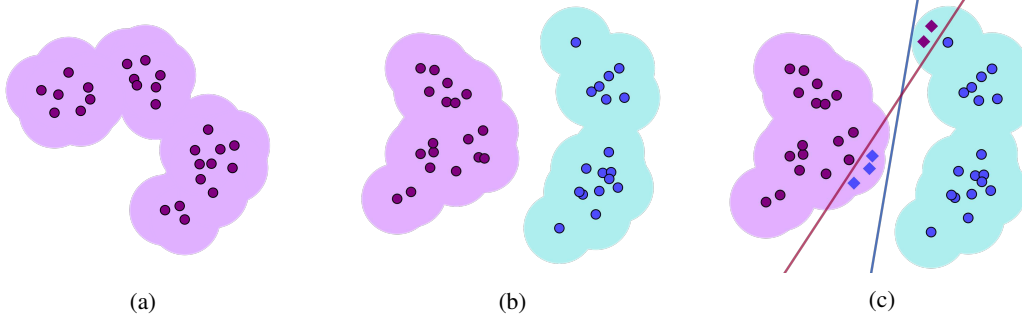

Figure 1: (a) The support of the distribution is such that $\mathsf{R}_\rho^C(h,c) = 0$ can only be achieved if $c$ is constant. (b) The $\rho$-expansion of the support of the distribution and target $c$ admit hypotheses $h$ such that $\mathsf{R}_\rho^C(h,c) = 0$. (c) An example where $\mathsf{R}_\rho^C$ and $\mathsf{R}_\rho^E$ differ. The red concept is the target, while the blue one is the hypothesis. The dots are the support of the distribution and the shaded regions represent their $\rho$-expansion. The diamonds represent perturbed inputs which cause $\mathsf{R}_\rho^E > 0$.

norm; in our case, the only meaningful norm is the Hamming distance. For a point $x \in \mathcal{X}$, let $B_\rho(x)$ denote the Hamming ball of radius $\rho$ around $x$. Given a distribution $D$ on $\mathcal{X}$, we consider the *adversarial risk* of a hypothesis $h$ with respect to a target concept $c$ and perturbation budget $\rho$. We focus on two definitions of risk. The *exact in the ball* risk $\mathsf{R}_\rho^E(h,c)$ is the probability $\mathbb{P}_{x \sim D} (\exists y \in B_\rho(x) \cdot h(y) \neq c(y))$ that the adversary can perturb a point $x$ drawn from distribution $D$ to a point $y$ such that $h(y) \neq c(y)$. The *constant in the ball* risk $\mathsf{R}_\rho^C(h,c)$ is the probability $\mathbb{P}_{x \sim D} (\exists y \in B_\rho(x) \cdot h(y) \neq c(x))$ that the adversary can perturb a point $x$ drawn from distribution $D$ to a point $y$ such that $h(y) \neq c(x)$. These definitions encode two different interpretations of robustness. In the first view, robustness speaks about the fidelity of the hypothesis to the target concept, whereas in the latter view robustness concerns the sensitivity of the output of the hypothesis to corruptions of the input. In fact, the latter view of robustness can in some circumstances be in conflict with accuracy in the traditional sense [26].

## 1.1 Overview of Our Contributions

We view our conceptual contributions to be at least as important as the technical results and believe that the issues highlighted in our work will result in more concrete theoretical frameworks being developed to study adversarial learning.

**Impossibility of Robust Learning in Distribution-Free PAC Setting**

We first consider the question of whether achieving *zero* (or low) robust risk is possible under either of the two definitions. If the *balls* of radius $\rho$ around the data points intersect so that the total region is connected, then unless the target function is constant, it is impossible to achieve $\mathsf{R}_\rho^C(h,c) = 0$ (see Figure 1). In particular, in most cases $\mathsf{R}_\rho^C(c,c) \neq 0$, i.e., even the target concept does not have zero risk with respect to itself. We show that this is the case for extremely simple concept classes such as *dictators* or *parities*. When considering the *exact on the ball* notion of robust learning, we at least have $\mathsf{R}_\rho^E(c,c) = 0$; in particular, any concept class that can be exactly learned can be robustly learned in this sense. However, even in this case we show that no "non-trivial" class of functions can be robustly learned. We highlight that these results show that a polynomial-size sample from the unknown distribution is not sufficient, even if the learning algorithm has arbitrary computational power (in the sense of Turing computability).[2]

**Robust Learning of Monotone Conjunctions**

Given the impossibility of distribution-free robust learning, we consider robust learning under specific distributions. We consider one of the simplest concept class studied in PAC Learning, the class of *monotone conjunctions*, under the class of $\log$-Lipschitz distributions (which includes the uniform distribution) and show that this class of functions is robustly learnable provided $\rho = O(\log n)$ and is not robustly learnable with polynomial sample complexity for $\rho = \omega(\log n)$. A class of distributions is said to be $\alpha$-$\log$-Lipschitz if the logarithm of the density function is $\log(\alpha)$-Lipschitz with respect to the Hamming distance. Our results apply in the setting where the learning algorithm only receives random labeled examples. On the other hand, a more powerful learning algorithm that has access to membership queries can exactly learn monotone conjunctions and as a result can also robustly learn with respect to *exact in the ball* loss.

**Computational Hardness of PAC Learning**

Finally, we consider computational aspects of robust learning. Our focus is on two questions: *computability* and *computational complexity*. Recent work by Bubeck et al. [7] provides a result that states that minimizing the robust loss on a polynomial-size sample suffices for robust learning. However, because of the existential quantifier over the ball implicit in the definition of the *exact in the ball* loss, the empirical risk cannot be *computed* as this requires enumeration over the *reals*. Even if one restricted attention to concepts defined over $\mathbb{Q}^n$, computing the loss would be *recursively enumerable*, but not *recursive*. In the case of functions defined over finite instance spaces, such as the boolean hypercube, the loss can be evaluated provided the learning algorithm has access to a membership query oracle; for the *constant in the ball* loss membership queries are not required. For functions defined on $\mathbb{R}^n$ it is unclear how either loss function can be evaluated even if the learner has access to membership queries, since in principle it requires enumerating over the reals. Under strong assumptions of *inductive bias* on the target and hypothesis class, it may be possible to evaluate the loss functions; however this would have to be handled on a case by case basis – for example, properties of the target and hypothesis, such as Lipschitzness or large margin, could be used to compute the exact in the ball loss in finite time.

Second, we consider the computational complexity of robust learning. Bubeck et al. [6] and Degwekar and Vaikuntanathan [9] have shown that there are concept classes that are hard to robustly learn under cryptographic assumptions, even when robust learning is information-theoretically feasible. Bubeck et al. [7] establish super-polynomial lower bounds for robust learning in the *statistical query* framework. We give an arguably simpler proof of hardness, based simply on the assumption that there exist concept classes that are hard to PAC learn. In particular, our reduction also implies that robust learning is hard even if the learning algorithm is allowed membership queries, provided the concept class that we reduce from is hard to learn using membership queries. Since the existence of one-way functions implies the existence of concept classes that are hard to PAC learn (with or without membership queries), our result is also based on a slightly weaker assumption than Bubeck et al. [7][3].

## 1.2 Related work on the Existence of Adversarial Examples

There is a considerable body of work that studies the inevitability of adversarial examples, e.g., [12, 14, 13, 16, 24]. These papers characterize robustness in the sense that a classifier's output on a point should not change if a perturbation of a certain magnitude is applied to it. Among other things, these works study geometrical characteristics of classifiers and statistical characteristics of classification data that lead to adversarial vulnerability.

Closer to the present paper are [10, 21, 22], which work the with exact-in-a-ball notion of robust risk. In particular, [10] considers the robustness of monotone conjunctions under the uniform distribution on the boolean hypercube for this notion of risk (therein called the *error region* risk). However [10] does not address the sample and computational complexity of learning: their results rather concern the ability of an adversary to magnify the missclassification error of *any* hypothesis with respect to *any* target function by perturbing the input. For example, they show that an adversary who can perturb $O(\sqrt{n})$ bits can increase the missclassification probability from $0.01$ to $1/2$. By contrast we show

that a weaker adversary, who can perturb only $\omega(\log n)$ bits, renders it impossible to learn monotone conjunctions with polynomial sample complexity. The main tool used in [10] is the isoperimetric inequality for the Boolean hypercube, which gives lower bounds on the volume of the expansions of arbitrary subsets. On the other hand, we use the probabilistic method to establish the existence of a single hard-to-learn target concept for any given algorithm with polynomial sample complexity.

## 2  Definition of Robust Learning

The notion of robustness can be accommodated within the basic set-up of PAC learning by adapting the definition of risk function. In this section we review two of the main definitions of *robust risk* that have been used in the literature. For concreteness we consider an input space $\mathcal{X} = \{0,1\}^n$ with metric $d : \mathcal{X} \times \mathcal{X} \to \mathbb{N}$, where $d(x,y)$ is the Hamming distance of $x, y \in \mathcal{X}$. Given $x \in \mathcal{X}$, we write $B_\rho(x)$ for the ball $\{y \in \mathcal{X} : d(x,y) \leq \rho\}$ with centre $x$ and radius $\rho \geq 0$.

The first definition of robust risk asks that the hypothesis be exactly equal to the target concept in the ball $B_\rho(x)$ of radius $\rho$ around a "test point" $x \in \mathcal{X}$:

**Definition 1.** *Given respective hypothesis and target functions $h, c : \mathcal{X} \to \{0,1\}$, distribution $D$ on $\mathcal{X}$, and robustness parameter $\rho \geq 0$, we define the "exact in the ball" robust risk of $h$ with respect to $c$ to be*

$$\mathsf{R}_\rho^E(h,c) = \mathop{\mathbb{P}}_{x \sim D}\left(\exists z \in B_\rho(x) : h(z) \neq c(z)\right) .$$

While this definition captures a natural notion of robustness, an obvious disadvantage is that evaluating the risk function requires the learner to have knowledge of the target function outside of the training set, e.g., through membership queries. Nonetheless, by considering a learner who has oracle access to the predicate $\exists z \in B_\rho(x) : h(z) \neq c(z)$, we can use the exact-in-the-ball framework to analyse sample complexity and to prove strong lower bounds on the computational complexity of robust learning.

A popular alternative to the exact-in-the-ball risk function in Definition 1 is the following *constant-in-the-ball risk* function:

**Definition 2.** *Given respective hypothesis and target functions $h, c : \mathcal{X} \to \{0,1\}$, distribution $D$ on $\mathcal{X}$, and robustness parameter $\rho \geq 0$, we define the "constant in the ball" robust risk of $h$ with respect to $c$ as*

$$\mathsf{R}_\rho^C(h,c) = \mathop{\mathbb{P}}_{x \sim D}\left(\exists z \in B_\rho(x) : h(z) \neq c(x)\right) .$$

An obvious advantage of the constant in the ball risk over the exact in the ball version is that in the former, evaluating the loss at point $x \in \mathcal{X}$ requires only knowledge of the correct label of $x$ and the hypothesis $h$. In particular, this definition can also be carried over to the non-realizable setting, in which there is no target. However, from a foundational point of view the constant in the ball risk has some drawbacks: recall from the previous section that under this definition it is possible to have strictly positive robust risk in the case that $h = c$. (Let us note in passing that the risk functions $\mathsf{R}_\rho^C$ and $\mathsf{R}_\rho^E$ are in general incomparable. Figure 1c gives an example in which $\mathsf{R}_\rho^C = 0$ and $\mathsf{R}_\rho^E > 0$.) Additionally, when we work in the hypercube, or a bounded input space, as $\rho$ becomes larger, we eventually require the function to be constant in the whole space. Essentially, to $\rho$-robustly learn in the realisable setting, we require concept and distribution pairs to be represented as two sets $D_+$ and $D_-$ whose $\rho$-expansions don't intersect, as illustrated in Figures 1a and 1b. These limitations appear even more stringent when we consider simple concept classes such as parity functions, which are defined for an index set $I \subseteq [n]$ as $f_I(x) = \sum_i x_i + b \mod 2$ for $b \in \{0,1\}$. This class can be PAC-learned, as well as exactly learned with $n$ membership queries. However, for any point, it suffices to flip one bit of the index set to switch the label, so $\mathsf{R}_\rho^C(f_I, f_I) = 1$ for any $\rho \geq 1$ if $I \neq \emptyset$.

Ultimately, we want the adversary's power to come from creating perturbations that cause the hypothesis and target functions to differ in some regions of the input space. For this reason we favor the exact-in-the-ball definition and henceforth work with that.

Having settled on a risk function, we now formulate the definition of robust learning. For our purposes a *concept class* is a family $\mathcal{C} = \{\mathcal{C}_n\}_{n \in \mathbb{N}}$, with $\mathcal{C}_n$ a class of functions from $\{0,1\}^n$ to $\{0,1\}$. Likewise a *distribution class* is a family $\mathcal{D} = \{\mathcal{D}_n\}_{n \in \mathbb{N}}$, with $\mathcal{D}_n$ a set of distributions on $\{0,1\}^n$. Finally a *robustness function* is a function $\rho : \mathbb{N} \to \mathbb{N}$.

**Definition 3.** *Fix a function $\rho : \mathbb{N} \to \mathbb{N}$. We say that an algorithm $\mathcal{A}$ efficiently $\rho$-robustly learns a concept class $\mathcal{C}$ with respect to distribution class $\mathcal{D}$ if there exists a polynomial $\mathrm{poly}(\cdot, \cdot, \cdot)$ such that for all $n \in \mathbb{N}$, all target concepts $c \in \mathcal{C}_n$, all distributions $D \in \mathcal{D}_n$, and all accuracy and confidence parameters $\epsilon, \delta > 0$, there exists $m \leq \mathrm{poly}(1/\epsilon, 1/\delta, n)$, such that when $\mathcal{A}$ is given access to a sample $S \sim D^m$ it outputs $h : \{0,1\}^n \to \{0,1\}$ such that $\underset{S \sim D^m}{\mathbb{P}} \left( \mathsf{R}^E_{\rho(n)}(h,c) < \epsilon \right) > 1 - \delta$.*

Note that the definition of robust learning requires polynomial sample complexity and allows improper learning (the hypothesis $h$ need not belong to the concept class $\mathcal{C}_n$).

In the standard PAC framework, a hypothesis $h$ is considered to have zero risk with respect to a target concept $c$ when $\underset{x \sim D}{\mathbb{P}} (h(x) \neq c(x)) = 0$. We have remarked that exact learnability implies robust learnability; we next give an example of a concept class $\mathcal{C}$ and distribution $D$ such that $\mathcal{C}$ is PAC learnable under $D$ with zero risk and yet cannot be robustly learned under $D$ (regardless of the sample complexity).

**Lemma 4.** *The class of dictators is not 1-robustly learnable (and thus not robustly learnable for any $\rho \geq 1$) with respect to the robust risk of Definition 1 in the distribution-free setting.*

*Proof.* Let $c_1$ and $c_2$ be the dictators on variables $x_1$ and $x_2$, respectively. Let $D$ be such that $\underset{x \sim D}{\mathbb{P}} (x_1 = x_2) = 1$ and $\underset{x \sim D}{\mathbb{P}} (x_k = 1) = \frac{1}{2}$ for $k \geq 3$. Draw a sample $S \sim D^m$ and label it according to $c \sim U(c_1, c_2)$. By the choice of $D$, the elements of $S$ will have the same label regardless of whether $c_1$ or $c_2$ was picked. However, for $x \sim D$, it suffices to flip any of the first two bits to cause $c_1$ and $c_2$ to disagree on the perturbed input. We can easily show that, for any $h \in \{0,1\}^{\mathcal{X}}$, $\mathsf{R}^E_1(c_1, h) + \mathsf{R}^E_1(c_2, h) \geq \mathsf{R}^E_1(c_1, c_2) = 1$. Then

$$\underset{c \sim U(c_1, c_2)}{\mathbb{E}} \underset{S \sim D^m}{\mathbb{E}} \left[ \mathsf{R}^E_1(h, c) \right] \geq 1/2 .$$

We conclude that one of $c_1$ or $c_2$ has robust risk at least 1/2. $\qquad\square$

Note that a PAC learning algorithm with error probability threshold $\varepsilon = 1/3$ will either output $c_1$ or $c_2$ and will hence have standard risk zero. We refer the reader to Appendix B for further discussion on the relationship between robust and zero-risk learning.

## 3 No Distribution-Free Robust Learning in $\{0,1\}^n$

In this section, we show that no non-trivial concept class is efficiently 1-robustly learnable in the boolean hypercube. Such a class is thus not efficiently $\rho$-robustly learnable for any $\rho \geq 1$. Efficient robust learnability then requires access to a more powerful learning model or distributional assumptions.

Let $\mathcal{C}_n$ be a concept class on $\{0,1\}^n$, and define a concept class as $\mathcal{C} = \bigcup_{n \geq 1} \mathcal{C}_n$. We say that a class of functions is trivial if $\mathcal{C}_n$ has at most two functions, and that they differ on every point.

**Theorem 5.** *Any concept class $\mathcal{C}$ is efficiently distribution-free robustly learnable iff it is trivial.*

The proof of the theorem relies on the following lemma:

**Lemma 6.** *Let $c_1, c_2 \in \{0,1\}^{\mathcal{X}}$ and fix a distribution on $\mathcal{X}$. Then for all $h : \{0,1\}^n \to \{0,1\}$*

$$\mathsf{R}^E_\rho(c_1, c_2) \leq \mathsf{R}^E_\rho(c_1, h) + \mathsf{R}^E_\rho(c_2, h) .$$

*Proof.* Let $x \in \{0,1\}^n$ be arbitrary, and suppose that $c_1$ and $c_2$ differ on some $z \in B_\rho(x)$. Then either $h(z) \neq c_1(z)$ or $h(z) \neq c_2(z)$. The result follows. $\qquad\square$

The idea of the proof of Theorem 5 (which can be found in Appendix C) is a generalization of the proof of Lemma 4 that dictators are not robustly learnable. However, note that we construct a distribution whose support is all of $\mathcal{X}$. It is possible to find two hypotheses $c_1$ and $c_2$ and create a distribution such that $c_1$ and $c_2$ will likely look identical on samples of size polynomial in $n$ but have robust risk $\Omega(1)$ with respect to one another. Since any hypothesis $h$ in $\{0,1\}^{\mathcal{X}}$ will disagree either

with $c_1$ or $c_2$ on a given point $x$ if $c_1(x) \neq c_2(x)$, by choosing the target hypothesis $c$ at random from $c_1$ and $c_2$, we can guarantee that $h$ won't be robust against $c$ with positive probability. Finally, note that an analogous argument can be made for a more general setting (for example in $\mathbb{R}^n$).

# 4    Monotone Conjunctions

It turns out that we do not need recourse to "bad" distributions to show that very simple classes of functions are not efficiently robustly learnable. As we demonstrate in this section, MON-CONJ, the class of monotone conjunctions, is not efficiently robustly learnable *even under the uniform distribution* for robustness parameters that are superlogarithmic in the input dimension.

## 4.1    Non-Robust Learnability

The idea to show that MON-CONJ is not efficiently robustly learnable is in the same vein as the proof of Theorem 5. We first start by proving the following lemma, which lower bounds the robust risk of two disjoint monotone conjunctions.

**Lemma 7.** *Under the uniform distribution, for any $n \in \mathbb{N}$, disjoint $c_1, c_2 \in$ MON-CONJ of length $3 \leq l \leq n/2$ on $\{0, 1\}^n$ and robustness parameter $\rho \geq l/2$, we have that $\mathsf{R}_\rho^E(c_1, c_2)$ is bounded below by a constant that can be made arbitrarily close to $\frac{1}{2}$ as $l$ gets larger.*

*Proof.* For a hypothesis $c \in$ MON-CONJ , let $I_c$ be the set of variables in $c$. Let $c_1, c_2 \in \mathcal{C}$ be as in the theorem statement. Then the robust risk $\mathsf{R}_\rho^E(c_1, c_2)$ is bounded below by

$$\mathbb{P}_{x \sim D}\left(c_1(x) = 0 \land x \text{ has at least } l/2 \text{ 1's in } I_{c_2}\right) = (1 - 2^{-l})/2 \ .$$

$\square$

Now, the following lemma shows that if we choose the length of the conjunctions $c_1$ and $c_2$ to be super-logarithmic in $n$, then, for a sample of size polynomial in $n$, $c_1$ and $c_2$ will agree on $S$ with probability at least $1/2$. The proof can be found in Appendix D.1.

**Lemma 8.** *For any functions $l(n) = \omega(\log(n))$ and $m(n) = poly(n)$, for any disjoint monotone conjunctions $c_1, c_2$ such that $|I_{c_1}| = |I_{c_2}| = l(n)$, there exists $n_0$ such that for all $n \geq n_0$, a sample $S$ of size $m(n)$ sampled i.i.d. from $D$ will have that $c_1(x) = c_2(x) = 0$ for all $x \in S$ with probability at least $1/2$.*

We are now ready to prove our main result of the section.

**Theorem 9.** *MON-CONJ is not efficiently $\rho$-robustly learnable for $\rho(n) = \omega(\log(n))$ under the uniform distribution.*

*Proof.* Fix any algorithm $\mathcal{A}$ for learning MON-CONJ . We will show that the expected robust risk between a randomly chosen target function and any hypothesis returned by $\mathcal{A}$ is bounded below by a constant. Fix a function $poly(\cdot, \cdot, \cdot, \cdot, \cdot)$, and note that, since $size(c)$ and $\rho$ are both at most $n$, we can simply consider a function $poly(\cdot, \cdot, \cdot)$ in the variables $1/\epsilon$, and $1/\delta, n$ instead. Let $\delta = 1/2$, and fix a function $l(n) = \omega(\log(n))$ that satisfies $l(n) \leq n/2$, and let $\rho(n) = l(n)/2$ ($n$ is not yet fixed). Let $n_0$ be as in Lemma 8, where $m(n)$ is the fixed sample complexity function. Then Equation (8) holds for all $n \geq n_0$.

Now, let $D$ be the uniform distribution on $\{0, 1\}^n$ for $n \geq \max(n_0, 3)$, and choose $c_1, c_2$ as in Lemma 7. Note that $\mathsf{R}_\rho^E(c_1, c_2) > \frac{5}{12}$ by the choice of $n$. Pick the target function $c$ uniformly at random between $c_1$ and $c_2$, and label $S \sim D^m$ with $c$, where $m = poly(1/\epsilon, 1/\delta, n)$. By Lemma 8, $c_1$ and $c_2$ agree with the labeling of $S$ (which implies that all the points have label 0) with probability at least $\frac{1}{2}$ over the choice of $S$.

Define the following three events for $S \sim D^m$:

$$\mathcal{E} : \ c_{1|S} = c_{2|S} \ , \quad \mathcal{E}_{c_1} : \ c = c_1 \ , \quad \mathcal{E}_{c_2} : \ c = c_2 \ .$$

Then, by Lemmas 8 and 6,

$$\mathop{\mathbb{E}}_{c,S}\left[\mathsf{R}_\rho^E(\mathcal{A}(S),c)\right] \geq \mathop{\mathbb{P}}_{c,S}(\mathcal{E}) \mathop{\mathbb{E}}_{c,S}\left[\mathsf{R}_\rho^E(\mathcal{A}(S),c) \mid \mathcal{E}\right]$$

$$> \frac{1}{2}\left(\mathop{\mathbb{P}}_{c,S}(\mathcal{E}_{c_1})\mathop{\mathbb{E}}_{S}\left[\mathsf{R}_\rho^E(\mathcal{A}(S),c) \mid \mathcal{E} \cap \mathcal{E}_{c_1}\right] + \mathop{\mathbb{P}}_{c,S}(\mathcal{E}_{c_2})\mathop{\mathbb{E}}_{S}\left[\mathsf{R}_\rho^E(\mathcal{A}(S),c) \mid \mathcal{E} \cap \mathcal{E}_{c_2}\right]\right)$$

$$= \frac{1}{4}\mathop{\mathbb{E}}_{S}\left[\mathsf{R}_\rho^E(\mathcal{A}(S),c_1) + \mathsf{R}_\rho^E(\mathcal{A}(S),c_2) \mid \mathcal{E}\right]$$

$$\geq \frac{1}{4}\mathop{\mathbb{E}}_{S}\left[\mathsf{R}_\rho^E(c_2,c_1)\right]$$

$$> 0.1 \ .$$

<div align="right">□</div>

## 4.2 Robust Learnability Against a Logarithmically-Bounded Adversary

The argument showing the non-robust learnability of MON-CONJ under the uniform distribution in the previous section cannot be carried through if the conjunction lengths are logarithmic in the input dimension, or if the robustness parameter is small compared to that target conjunction's length. In both cases, we show that it is possible to efficiently robustly learn these conjunctions if the class of distributions is $\alpha$-log-Lipschitz, i.e. there exists a universal constant $\alpha \geq 1$ such that for all $n \in \mathbb{N}$, all distributions $D$ on $\{0,1\}^n$ and for all input points $x, x' \in \{0,1\}^n$, if $d_H(x,x') = 1$, then $|\log(D(x)) - \log(D(x'))| \leq \log(\alpha)$ (see Appendix A.3 for further details and useful facts).

**Theorem 10.** *Let $\mathcal{D} = \{\mathcal{D}_n\}_{n \in \mathbb{N}}$, where $\mathcal{D}_n$ is a set of $\alpha$-log-Lipschitz distributions on $\{0,1\}^n$ for all $n \in \mathbb{N}$. Then the class of monotone conjunctions is $\rho$-robustly learnable with respect to $\mathcal{D}$ for robustness function $\rho(n) = O(\log n)$.*

The proof can be found in Appendix D. This combined with Theorem 10 shows that $\rho(n) = \log(n)$ is essentially the threshold for efficient robust learnability of the class MON-CONJ .

## 5 Computational Hardness of Robust Learning

In this section, we establish that the computational hardness of PAC-learning a concept class $\mathcal{C}$ with respect to a distribution class $\mathcal{D}$ implies the computational hardness of robustly learning a family of concept-distribution pairs from a related class $\mathcal{C}'$ and a restricted class of distributions $\mathcal{D}'$. This is essentially a version of the main result of [7], which used the constant-in-the-ball definition of robust risk. Our proof also uses the [7] trick of encoding a point's label in the input for the robust learning problem. Interestingly, our proof does not rely on any assumption other than the existence of a hard learning problem in the PAC framework and is valid under *both* Definitions 1 and 2 of robust risk.

**Construction of $\mathcal{C}'$.** Suppose we are given $\mathcal{C} = \{\mathcal{C}_n\}_{n \in \mathbb{N}}$ and $\mathcal{D} = \{\mathcal{D}_n\}_{n \in \mathbb{N}}$ with $\mathcal{C}_n$ and $\mathcal{D}_n$ defined on $\mathcal{X}_n = \{0,1\}^n$. Given $k \in \mathbb{N}$, we define the family of concept and distribution pairs $\{(c',D')\}_{D' \in \mathcal{D}'_{c'}, c' \in \mathcal{C}'}$, where $\mathcal{C}' = \{\mathcal{C}'_{(k,n)}\}_{k,n \in \mathbb{N}}$ on $\mathcal{X}'_{k,n} = \{0,1\}^{(2k+1)n+1}$ as follows. Let $\mathrm{maj}_k : \mathcal{X}'_{k,n} \to \mathcal{X}_n$ be the function that returns the majority vote on each subsequent block of $k$ bits, and ignores the last bit. We define $\mathcal{C}'_{(k,n)} = \{c \circ \mathrm{maj}_{2k+1} \mid c \in \mathcal{C}_n\}$. Let $\varphi_k : \mathcal{X}_n \to \mathcal{X}'_{k,n}$ be defined as

$$\varphi_k(x) := \underbrace{x_1 \ldots x_1 x_2 \ldots x_{d-1} x_d \ldots x_d}_{2k+1 \text{ copies of each } x_i} c(x) \ , \quad \varphi_k(S) := \{\varphi_k(x_i) \mid x_i \in S\} \ ,$$

for $x = x_1 x_2 \ldots x_d \in \mathcal{X}$ and $S \subseteq \mathcal{X}$. For a concept $c \in \mathcal{C}_n$, each $D \in \mathcal{D}_n$ induces a distribution $D' \in \mathcal{D}'_{c'}$, where $c' = c \circ \mathrm{maj}_{2k+1}$ and $D'(z) = D(x)$ if $z = \varphi_k(x)$, and $D'(z) = 0$ otherwise.

As shown below, this set up allows us to see that any algorithm for learning $\mathcal{C}_n$ with respect to $\mathcal{D}_n$ yields an algorithm for learning the pairs $\{(c',D')\}_{D' \in \mathcal{D}'_{c'}, c' \in \mathcal{C}'}$. However, any *robust* learning algorithm cannot solely rely on the last bit of the input, as it could be flipped by an adversary. Then, this algorithm can be used to PAC-learn $\mathcal{C}_n$. This establishes the equivalence of the computational difficulty between PAC-learning $\mathcal{C}_n$ with respect to $\mathcal{D}_n$ and robustly learning $\{(c',D')\}_{D' \in \mathcal{D}'_{c'}, c' \in \mathcal{C}'_{(k,n)}}$.

As mentioned earlier, we can still efficiently PAC-learn the pairs $\{(c', D')\}_{D' \in \mathcal{D}'_{c'}, c' \in \mathcal{C}'}$ simply by always outputting a hypothesis that returns the last bit of the input.

**Theorem 11.** *For any concept class $\mathcal{C}_n$, family of distributions $\mathcal{D}_n$ over $\{0,1\}^n$ and $k \in \mathbb{N}$, there exists a concept class $\mathcal{C}'_{(k,n)}$ and a family of distributions $\mathcal{D}'_{(k,n)}$ over $\{0,1\}^{(2k+1)n+1}$ such that efficient $k$-robust learnability of the concept-distribution pairs $\{(c', D')\}_{D' \in \mathcal{D}'_{c'}, c' \in \mathcal{C}'_{(k,n)}}$ and either of the robust risk functions $\mathsf{R}_k^C$ or $\mathsf{R}_k^E$ implies efficient PAC-learnability of $\mathcal{C}_n$ with respect to $\mathcal{D}_n$.*

Before proving the above result, let us first prove the following proposition.

**Proposition 12.** *The concept-distribution pairs $\{(c', D')\}_{D' \in \mathcal{D}'_{c'}, c' \in \mathcal{C}'_{(k,n)}}$ can be $k$-robustly learned using $O\left(\frac{1}{\epsilon}\left(\log |\mathcal{C}_n| + \log \frac{1}{\delta}\right)\right)$ examples.*

*Proof.* First note that, since $\mathcal{C}_n$ is finite, we can use PAC-learning sample bounds for the realizable setting (see for example [23]) to get that the sample complexity of learning $\mathcal{C}_n$ is $O\left(\frac{1}{\epsilon}(\log |\mathcal{C}_n| + \log \frac{1}{\delta})\right)$. Now, if we have PAC-learned $\mathcal{C}_n$ with respect to $\mathcal{D}_n$, and $h$ is the hypothesis returned on a sample labeled according to a target concept $c \in \mathcal{C}_n$, we can compose it with the function $\text{maj}_k$ to get a hypothesis $h'$ for which any perturbation of at most $k$ bits of $x' \sim D'$ (where $D'$ is the distribution induced by the target concept $c$ and distribution $D$) will not change $h'(x')$. Thus, we also have $k$-robustly learned $\mathcal{C}'_{(k,n)}$. $\square$

*Remark* 13. The sample complexity in Proposition 12 is independent of $k$, and so the construction of the class $\mathcal{C}'$ on $\mathcal{X}'$ allows the adversary to modify $\frac{1}{2n}$ fraction of the bits. There are ways to make the adversary more powerful and keep the sample complexity unchanged. Indeed, the fraction of the bits the adversary can flip can be increased by using error correction codes. For example, BCH codes [5, 17] would allow us to obtain an input space $\mathcal{X}'$ of dimension $n + k \log n$ where the adversary can flip $\frac{k}{n + k \log n}$ bits.

We are now ready to prove the main result of this section.

*Proof of Theorem 11.* Given $\mathcal{C}_n$ and $\mathcal{D}$, let $\mathcal{C}'_{(k,n)}$ and $\{\mathcal{D}'_{c'}\}_{c' \in \mathcal{C}'_{(k,n)}}$ be constructed as above. Suppose that it is hard to PAC-learn $\mathcal{C}_n$ with respect to the distribution family $\mathcal{D}_n$. Suppose that we are given an algorithm $\mathcal{A}'$ to $k$-robustly learn $\{(c', D')\}_{D' \in \mathcal{D}'_{c'}, c' \in \mathcal{C}'_{(k,n)}}$ and a sample complexity $m$.

Let $\epsilon, \delta > 0$ be arbitrary and $c \in \mathcal{C}_n$ be an arbitrary target concept and let $c' \in \mathcal{C}'_{(k,n)}$ be such that $c' = c \circ \text{maj}_{2k+1}$. Let $D \in \mathcal{D}_n$ be a distribution on $\mathcal{X}_n$, and let $D' \in \mathcal{D}'_{c'}$ be its induced distribution on $\mathcal{X}'_{k,n}$. A PAC-learning algorithm for $\mathcal{C}_n$ is as follows. Draw a sample $S \sim D^m$ and let $S' = \varphi_k(S)$. Note that this simulates a sample $S' \sim D'^m$, and that $c'$ will give the same label to all points in the $\rho$-ball centred at $x'$ for any $x'$ in the support of $D'$.

Since $\mathcal{A}'$ $k$-robustly learns the concept-distribution pairs $\{(c', D')\}_{D' \in \mathcal{D}'_{c'}, c' \in \mathcal{C}'_{(k,n)}}$, with probability at least $1 - \delta$ over $S'$, for any $x \sim D$, we have that $h'$ will be wrong on $\varphi_k(x)$ (where the last bit is random) with probability at most $\epsilon$. So by outputting $h = h' \circ \varphi_k$, we have an algorithm to PAC-learn $\mathcal{C}_n$ with respect to the distribution family $\mathcal{D}_n$. $\square$

# 6   Conclusion

We have studied robust learnability from a computational learning theory perspective and have shown that efficient robust learning can be hard – even in very natural and apparently straightforward settings. We have moreover given a tight characterization of the strength of an adversary to prevent robust learning of monotone conjunctions under certain distributional assumptions. An interesting avenue for future work is to see whether this result can be generalised to other classes of functions. Finally, we have provided a simpler proof of the previously established result of the computational hardness of robust learning.

In the light of our results, it seems to us that more thought needs to be put into what we want out of robust learning in terms of computational efficiency and sample complexity, which will inform our choice of risk functions. Indeed, at first glance, robust learning definitions that have appeared in prior work seem in many ways natural and reasonable; however, their inadequacies surface when viewed

under the lens of computational learning theory. Given our negative results in the context of the current robustness models, one may surmise that requiring a classifier to be correct in an entire ball near a point is asking for too much. Under such a requirement, we can only solve "easy problems" with strong distributional assumptions. Nevertheless, it may still be of interest to study these notions of robust learning in different learning models, for example where one has access to membership queries.

### Acknowledgments

Varun Kanade was supported in part by the Alan Turing Institute under the EPSRC grant EP/N510129/1.

## Footnotes

[1]For an in-depth review and definitions of different types of attacks, the reader may refer to [3, 11].

[2]We do require any operation performed by the learning algorithm is computable; the results of Bubeck et al. [7] imply that an algorithm that can potentially evaluate *uncomputable* functions can always robustly learn using a polynomial-size sample. See the discussion on computational hardness below.

[3]It is believed that the existence of hard to PAC learn concept classes is not sufficient to construct one-way functions. [1].

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
