[Supplementary Material]

# On the Hardness of Robust Classification: Appendix

## A   Learning Theory Basics

### A.1   The PAC framework

We study the problem of robust classification. This is a generalization of standard classification tasks, which are defined on an input space $\mathcal{X}_n$ of dimension $n$ and finite output space $\mathcal{Y}$. Common examples of input spaces are $\{0, 1\}^n$, $[0, 1]^n$, and $\mathbb{R}^n$. We focus on *binary classification* in the *realizable setting*, where $\mathcal{Y} = \{0, 1\}$, and we get access to a sample $S = \{(x_i, y_i)\}_{i=1}^m$ where the $x_i$'s are drawn i.i.d. from an unknown underlying distribution $D$, and there exists $c : \mathcal{X} \to \mathcal{Y}$ such that $y_i = c(x_i)$, namely, there exists a *target concept* that has labeled the sample. In the PAC framework [27], our goal is to find a function $h$ that approximates $c$ with high probability over the training sample. This means we are allowing a small chance of having a sample that is not representative of the distribution. As we require our confidence to increase, we require more data. PAC learning is formally defined for *concept classes* $\mathcal{C}_n \subseteq \{0, 1\}^{\mathcal{X}_n}$ as follows.

**Definition 14** (PAC Learning). *Let $\mathcal{C}_n$ be a concept class over $\mathcal{X}_n$ and let $\mathcal{C} = \bigcup_{n \in \mathbb{N}} \mathcal{C}_n$. We say that $\mathcal{C}$ is PAC learnable using hypothesis class $\mathcal{H}$ and sample complexity function $p(\cdot, \cdot, \cdot)$ if there exists an algorithm $\mathcal{A}$ that satisfies the following: for all $n \in \mathbb{N}$, for every $c \in \mathcal{C}_n$, for every $D$ over $\mathcal{X}_n$, for every $0 < \epsilon < 1/2$ and $0 < \delta < 1/2$, if whenever $\mathcal{A}$ is given access to $m \geq p(n, 1/\epsilon, 1/\delta)$ examples drawn i.i.d. from $D$ and labeled with $c$, $\mathcal{A}$ outputs $h \in \mathcal{H}$ such that with probability at least $1 - \delta$,*

$$\mathbb{P}_{x \sim D} (c(x) \neq h(x)) \leq \epsilon .$$

*We say that $\mathcal{C}$ is statistically efficiently PAC learnable if $p$ is polynomial in $n, 1/\epsilon$ and $1/\delta$, and computationally efficiently PAC learnable if $\mathcal{A}$ runs in polynomial time in $n, 1/\epsilon$ and $1/\delta$.*

PAC learning is *distribution-free*, in the sense that no assumptions are made about the distribution from which the data comes from. The setting where $\mathcal{C} = \mathcal{H}$ is called *proper learning*, and *improper learning* otherwise.

### A.2   Monotone Conjunctions

A conjunction $c$ over $\{0, 1\}^n$ can be represented a set of literals $l_1, \ldots, l_k$, where, for $x \in \mathcal{X}_n$, $c(x) = \bigwedge_{i=1}^k l_i$. For example, $c(x) = x_1 \wedge \bar{x_2} \wedge x_5$ is a conjunction. Monotone conjunctions are the subclass of conjunctions where negations are not allowed, i.e. all literals are of the form $l_i = x_j$ for some $j \in [n]$.

The standard PAC learning algorithm to learn monotone conjunctions is as follows. We start with the hypothesis $h(x) = \bigwedge_{i \in I_h} x_i$, where $I_h = [n]$. For each example $x$ in $S$, we remove $i$ from $I_h$ if $c(x) = 1$ and $x_i = 0$.

When one has access to membership queries, one can easily exactly learn monotone conjunctions over the whole input space: we start with the instance where all bits are 1 (which is always a positive example), and we can test whether each variable is in the target conjunction by setting the corresponding bit to 0 and requesting the label.

We refer the reader to [23] for an in-depth introduction to machine learning theory.

### A.3   Log-Lipschitz Distributions

**Definition 15.** *A distribution $D$ on $\{0, 1\}^n$ is said to be $\alpha$-log-Lipschitz if for all input points $x, x' \in \{0, 1\}^n$, if $d_H(x, x') = 1$, then $|\log(D(x)) - \log(D(x'))| \leq \log(\alpha)$.*

The intuition behind log-Lipschitz distributions is that points that are close to each other must not have frequencies that greatly differ from each other. Note that, by definition, $D(x) > 0$ for all inputs $x$. Moreover, the uniform distribution is log-Lipschitz with parameter $\alpha = 1$. Another example of log-Lipschitz distributions is the class of product distributions where the probability of drawing a 0 (or equivalently a 1) at index $i$ is in the interval $\left[\frac{1}{1+\alpha}, \frac{\alpha}{1+\alpha}\right]$. Log-Lipschitz distributions have been studied in [2], and its variants in [15, 18].

Log-Lipschitz distributions have the following useful properties, which we will often refer to in our proofs.

**Lemma 16.** *Let $D$ be an $\alpha$-log-Lipschitz distribution over $\{0,1\}^n$. Then the following hold:*

1. *For $b \in \{0,1\}$, $\frac{1}{1+\alpha} \leq \mathbb{P}_{x \sim D}(x_i = b) \leq \frac{\alpha}{1+\alpha}$.*

2. *For any $S \subseteq [n]$, the marginal distribution $D_{\bar{S}}$ is $\alpha$-log-Lipschitz, where $D_{\bar{S}}(y) = \sum_{y' \in \{0,1\}^S} D(yy')$.*

3. *For any $S \subseteq [n]$ and for any property $\pi_S$ that only depends on variables $x_S$, the marginal with respect to $\bar{S}$ of the conditional distribution $(D|\pi_S)_{\bar{S}}$ is $\alpha$-log-Lipschitz.*

4. *For any $S \subseteq [n]$ and $b_S \in \{0,1\}^S$, we have that $\left(\frac{1}{1+\alpha}\right)^{|S|} \leq \mathbb{P}_{x \sim D}(x_i = b) \leq \left(\frac{\alpha}{1+\alpha}\right)^{|S|}$.*

*Proof.* To prove (1), fix $i \in [n]$ and $b \in \{0,1\}$ and denote by $x^{\oplus i}$ the result of flipping the $i$-th bit of $x$. Note that

$$\mathbb{P}_{x \sim D}(x_i = b) = \sum_{\substack{z \in \{0,1\}^n: \\ z_i = b}} D(z) = \sum_{\substack{z \in \{0,1\}^n: \\ z_i = b}} \frac{D(z)}{D(z^{\oplus i})} D(z^{\oplus i}) \leq \alpha \sum_{\substack{z \in \{0,1\}^n: \\ z_i = b}} D(z^{\oplus i}) = \alpha \mathbb{P}_{x \sim D}(x_i \neq b) \ .$$

The result follows from solving for $\mathbb{P}_{x \sim D}(x_i = b)$.

Without loss of generality, let $\bar{S} = \{1, \ldots, k\}$ for some $k \leq n$. Let $x, x' \in \{0,1\}^{\bar{S}}$ with $d_H(x, x') = 1$.

To prove (2), let $D_{\bar{S}}$ be the marginal distribution. Then,

$$D_{\bar{S}}(x) = \sum_{y \in \{0,1\}^S} D(xy) = \sum_{y \in \{0,1\}^S} \frac{D(xy)}{D(x'y)} D(x'y) \leq \alpha \sum_{y \in \{0,1\}^S} D(x'y) = \alpha D_{\bar{S}}(x') \ .$$

To prove (3), denote by $X_{\pi_S}$ the set of points in $\{0,1\}^S$ satisfying property $\pi_S$, and by $xX_{\pi_S}$ the set of inputs of the form $xy$, where $y \in X_{\pi_S}$. By a slight abuse of notation, let $D(X_{\pi_S})$ be the probability of drawing a point in $\{0,1\}^n$ that satisfies $\pi_S$. Then,

$$D(xX_{\pi_S}) = \sum_{y \in X_{\pi_S}} D(xy) = \sum_{y \in X_{\pi_S}} \frac{D(xy)}{D(x'y)} D(x'y) \leq \alpha \sum_{y \in X_{\pi_S}} D(x'y) = \alpha D(x'X_{\pi_S}) \ .$$

We can use the above and show that

$$(D|\pi_S)_{\bar{S}}(x) = \frac{D(xX_{\pi_S})}{D(x'X_{\pi_S})} \frac{D(x'X_{\pi_S})}{D(X_{\pi_S})} \leq \alpha(D|\pi_S)_{\bar{S}}(x') \ .$$

Finally, (4) is a corollary of (1)–(3).

$\square$

# B Discussion on the Relationship between Robust and Zero-Risk Learning

We saw that, for both robust risks $\mathsf{R}_\rho^C$ and $\mathsf{R}_\rho^E$, zero-risk learning does not necessarily imply robust learning. Moreover, as shown in Section 3, efficient distribution-free robust learning is not possible even in the realizable setting. What can be said if we have access to a robust learning algorithm for a specific distribution on the boolean hypercube? We will show that distribution-dependent robust learning implies zero-risk learning for both robust risk definitions, under certain conditions on the measure of balls in the support of the distribution. Let us start with Definition 1, where we require the hypothesis to be exact in the $\rho$-balls around a point.

**Proposition 17.** *For any probability measure $\mu$ on $\{0,1\}^n$, robustness parameter $\rho$ and concepts $h, c$, there exists $\epsilon > 0$ such that if $\mathsf{R}_\rho^E(h, c) < \epsilon$ then $h(x) = c(x)$ for any $x \in \mathcal{X}$ such that $\mu(B_\rho(x)) > 0$. In particular, one has that $h$ and $c$ agree on the support of $\mu$.*

*Proof.* Suppose there exists $x^* \in \mathcal{X}$ with $\mu(B_\rho(x^*)) > 0$ such that $h(x^*) \neq c(x^*)$. Then for any $z \in B_\rho(x^*)$, we have that $\mathsf{R}_\rho^E(h, c, z)$, the robust risk of $h$ with respect to $c$ at point $z$, is 1. Let $\tilde{\mathcal{X}} := \{x \in \mathcal{X} : \mu(B_\rho(x)) > 0\}$, and $\epsilon = \min_{x \in \tilde{\mathcal{X}}} \mu(B_\rho(x))$. We have that

$$\mathsf{R}_\rho^E(h, c) \geq \sum_{z \in B_\rho(x^*)} \mu(\{z\}) \ell_\rho^R(h, c, z) = \mu(B_\rho(x^*)) \geq \epsilon \ .$$

□

**Corollary 18.** *For any fixed distribution $D$, robust learning with respect to $D$ implies zero-risk learning with respect to $D$ for any robustness parameter as long as $\epsilon$ in Proposition 17 satisfies $\epsilon^{-1} = poly(n)$.*

*Proof.* Fix a distribution $D \in \mathcal{D}$ on $\mathcal{X}$. Suppose that we have a $\rho$-robust learning algorithm $\mathcal{A}_{\mathcal{F}}^R(D)$ for $\mathcal{F}$, namely for all $\epsilon, \delta, \rho > 0$, for all $c \in \mathcal{F}$, if $\mathcal{A}_{\mathcal{F}}^R(D)$ has access to a sample $S$ of size $m \geq \text{poly}(\frac{1}{\epsilon}, \frac{1}{\delta}, \text{size}(c), n)$, it returns $f \in \mathcal{F}$ such that

$$\mathop{\mathbb{P}}_{S \sim D^m} \left( \ell_\rho^R(f, c) < \epsilon \right) \geq 1 - \delta \ . \tag{1}$$

By Proposition 17, we can choose $\epsilon$ such that $\mathsf{R}_\rho^E(h, c) < \epsilon$ implies that $h(x) = c(x)$ for any $x \in \mathcal{X}$ such that $\mu(B_\rho(x)) > 0$. Note that this $\epsilon$ depends on $D$, $\rho$ and $n$. So we have that

$$\mathop{\mathbb{P}}_{x \sim D} (f(x) \neq c(x)) = 0 \ , \tag{2}$$

with probability at least $1 - \delta$ over the training sample $S$, whose size remains polynomial in $\frac{1}{\delta}$ and $n$ by the proposition assumptions. □

*Remark* 19. The assumption on $\epsilon$ in Corollary 18 is necessary to use the robust learning algorithm as a black box: in Section 4.2, we work under a well-behaved class of distributions that includes the uniform distribution and show that, for long enough monotone conjunctions and small enough robustness parameter (with respect to the conjunction length), efficient robust learning is possible. However, we cannot exactly learn these monotone conjunctions. In the uniform distribution setting, the $\rho$-balls all have the same probability mass and $\epsilon^{-1}$ is essentially superpolynomial in $n$.

To show the same result for $\mathsf{R}_\rho^C$, where the hypothesis is constant in a ball, we can use the exact same reasoning as in Corollary 18, except that we need to show the analogue of Proposition 17 for this setting.

**Proposition 20.** *For any probability measure $\mu$ on $\{0,1\}^n$ and for any concepts $h, c$, there exists $\epsilon > 0$ such that if $\mathsf{R}_\rho^C(h, c) < \epsilon$ then $h$ and $c$ agree on the support of $\mu$.*

*Proof.* Fix $h, c, D$ and let $\epsilon = \min_{x \in \text{supp}(\mu)} \mu(\{x\})$. Suppose there exists $x^* \in \text{supp}(\mu)$ and $z \in B_\rho(x^*)$ such that $c(x^*) \neq h(z)$. Then

$$\mathsf{R}_\rho^C(h, c) = \mathop{\mathbb{P}}_{x \sim \mu} (\exists z \in B_\rho(x) \ . \ c(x) \neq h(z)) \geq \epsilon \ .$$

□

# C  Proofs from Section 3

*Proof of Theorem 5.* First, if $\mathcal{C}$ is trivial, we need at most one example to identify the target function.

For the other direction, suppose that $\mathcal{C}$ is non-trivial. We first start by fixing any learning algorithm and polynomial sample complexity function $m$. Let $\eta = \frac{1}{2^{\omega(\log n)}}$, $0 < \delta < \frac{1}{2}$, and note that for any constant $a > 0$,

$$\lim_{n \to \infty} n^a \log(1 - \eta)^{-1} = 0 \ ,$$

and so any polynomial in $n$ is $o\left(\left(\log(1/(1-\eta)))\right)^{-1}\right)$. Then it is possible to choose $n_0$ such that for all $n \geq n_0$,

$$m \leq \frac{\log(1/\delta)}{2n\log(1-\eta)^{-1}} \quad . \tag{3}$$

Since $\mathcal{C}$ is non-trivial, we can choose concepts $c_1, c_2 \in \mathcal{C}_n$ and points $x, x' \in \{0,1\}^n$ such that $c_1$ and $c_2$ agree on $x$ but disagree on $x'$. This implies that there exists a point $z \in \{0,1\}^n$ such that (i) $c_1(z) = c_2(z)$ and (ii) it suffices to change *only one bit* in $I := I_{c_1} \cup I_{c_2}$ to cause $c_1$ to disagree on $z$ and its perturbation. Let $D$ be such that

$$\mathbb{P}_{x \sim D}(x_i = z_i) = \begin{cases} 1-\eta & \text{if } i \in I \\ \frac{1}{2} & \text{otherwise} \end{cases} \quad .$$

Draw a sample $S \sim D^m$ and label it according to $c \sim U(c_1, c_2)$. Then,

$$\mathbb{P}_{S \sim D^m}(\forall x \in S \quad c_1(x) = c_2(x)) \geq (1-\eta)^{m|I|} \quad . \tag{4}$$

Bounding the RHS below by $\delta > 0$, we get that, as long as

$$m \leq \frac{\log(1/\delta)}{|I|\log(1-\eta)^{-1}} \quad , \tag{5}$$

(4) holds with probability at least $\delta$. But this is true as Equation (3) holds as well. However, if $x = z$, then it suffices to flip one bit of $x$ to get $x'$ such that $c_1(x') \neq c_2(x')$. Then,

$$\mathsf{R}_\rho^E(c_1, c_2) \geq \mathbb{P}_{x \sim D}(x_I = z_I) = (1-\eta)^{|I|} \quad . \tag{6}$$

The constraints on $\eta$ and the fact that $|I| \leq n$ are sufficient to guarantee that the RHS is $\Omega(1)$. Let $\alpha > 0$ be a constant such that $\mathsf{R}_\rho^E(c_1, c_2) \geq \alpha$.

We can use the same reasoning as in Lemma 6 to argue that, for any $h \in \{0,1\}^{\mathcal{X}}$,

$$\mathsf{R}_1^E(c_1, h) + \mathsf{R}_1^E(c_2, h) \geq \mathsf{R}_1^E(c_1, c_2) \quad .$$

Finally, we can show that

$$\mathbb{E}_{c \sim U(c_1, c_2)} \mathbb{E}_{S \sim D^m}\left[\mathsf{R}_1^R(h, c)\right] \geq \alpha\delta/2,$$

hence there exists a target $c$ with expected robust risk bounded below by a constant[4]. $\qquad\square$

## D  Proofs from Section 4

### D.1  Proof of Lemma 8

*Proof.* We begin by bounding the probability that $c_1$ and $c_2$ agree on an i.i.d. sample of size $m$:

$$\mathbb{P}_{S \sim D^m}(\forall x \in S \cdot c_1(x) = c_2(x) = 0) = \left(1 - \frac{1}{2^l}\right)^{2m} \quad . \tag{7}$$

Bounding the RHS below by $1/2$, we get that, as long as

$$m \leq \frac{\log(2)}{2\log(2^l/(2^l-1))} \quad , \tag{8}$$

(7) holds with probability at least $1/2$.

Now, if $l = \omega(\log(n))$, then for a constant $a > 0$,

$$\lim_{n \to \infty} n^a \log\left(\frac{2^l}{2^l - 1}\right) = 0 \quad ,$$

and so any polynomial in $n$ is $o\left(\left(\log\left(\frac{2^l}{2^l-1}\right)\right)^{-1}\right)$. $\qquad\square$

## D.2 Proof of Theorem 10

*Proof.* We show that the algorithm $\mathcal{A}$ for PAC-learning monotone conjunctions (see [23], chapter 2) is a robust learner for an appropriate choice of sample size. We start with the hypothesis $h(x) = \bigwedge_{i \in I_h} x_i$, where $I_h = [n]$. For each example $x$ in $S$, we remove $i$ from $I_h$ if $c(x) = 1$ and $x_i = 0$.

Let $\mathcal{D}$ be a class of $\alpha$-log-Lipschitz distributions. Let $n \in \mathbb{N}$ and $D \in \mathcal{D}_n$. Suppose moreover that the target concept $c$ is a conjunction of $l$ variables. Fix $\varepsilon, \delta > 0$. Let $\eta = \frac{1}{1+\alpha}$, and note that by Lemma 16, for any $S \subseteq [n]$ and $b_S \in \{0,1\}^S$, we have that $\eta^{|S|} \leq \mathbb{P}_{x \sim D} (x_i = b) \leq (1-\eta)^{|S|}$.

**Claim 1.** If $m \geq \left\lceil \frac{\log n - \log \delta}{\eta^{l+1}} \right\rceil$ then given a sample $S \sim D^m$, algorithm $\mathcal{A}$ outputs $c$ with probability at least $1 - \delta$.

*Proof of Claim 1.* Fix $i \in \{1, \ldots, n\}$. Algorithm $\mathcal{A}$ eliminates $i$ from the output hypothesis just in case there exists $x \in S$ with $x_i = 0$ and $c(x) = 1$. Now we have $\mathbb{P}_{x \sim D} (x_i = 0 \wedge c(x) = 1) \geq \eta^{l+1}$ and hence

$$\mathbb{P}_{S \sim D} (\forall x \in S \cdot i \text{ remains in } I_h) \leq (1 - \eta^{l+1})^m \leq e^{-m\eta^{l+1}} = \frac{\delta}{n} .$$

The claim now follows from union bound over $i \in \{1, \ldots, n\}$.

**Claim 2.** If $l \geq \frac{8}{\eta^2} \log(\frac{1}{\varepsilon})$ and $\rho \leq \frac{\eta l}{2}$ then $\mathbb{P}_{x \sim D} (\exists z \in B_\rho(x) \cdot c(z) = 1) \leq \varepsilon$.

*Proof of Claim 2.* Define a random variable $Y = \sum_{i \in I_c} \mathbb{I}(x_i = 1)$. We simulate $Y$ by the following process. Let $X_1, \ldots, X_l$ be random variables taking value in $\{0, 1\}$, and which may be dependent. Let $D_i$ be the marginal distribution on $X_i$ conditioned on $X_1, \ldots, X_{i-1}$. This distribution is also $\alpha$-log-Lipschitz by Lemma 16, and hence,

$$\mathbb{P}_{X_i \sim D_i} (X_i = 1) \leq 1 - \eta . \tag{9}$$

Since we are interested in the random variable $Y$ representing the number of 1's in $X_1, \ldots, X_l$, we define the random variables $Z_1, \ldots, Z_l$ as follows:

$$Z_k = \left( \sum_{i=1}^{k} X_i \right) - k(1 - \eta) .$$

The sequence $Z_1, \ldots, Z_l$ is a supermartingale with respect to $X_1, \ldots, X_l$:

$$\mathbb{E}\left[ Z_{k+1} \mid X_1, \ldots, X_k \right] = \mathbb{E}\left[ Z_k + X'_{k+1} - (1 - \eta) \mid X'_1, \ldots, X'_k \right]$$

$$= Z_k + \mathbb{P}\left( X'_{k+1} = 1 \mid X'_1, \ldots, X'_k \right) - (1 - \eta)$$

$$\leq Z_k . \tag{by (9)}$$

Now, note that all $Z_k$'s satisfy $|Z_{k+1} - Z_k| \leq 1$, and that $Z_l = Y - l(1 - \eta)$. We can thus apply the Azuma-Hoeffding (A.H.) Inequality to get

$$\mathbb{P}(Y \geq l - \rho) \leq \mathbb{P}\left( Y \geq l(1 - \eta) + \sqrt{2 \ln(1/\varepsilon)l} \right)$$

$$= \mathbb{P}\left( Z_l - Z_0 \geq \sqrt{2 \ln(1/\varepsilon)l} \right)$$

$$\leq \exp\left( -\frac{\sqrt{2 \ln(1/\varepsilon)l}^2}{2l} \right) \tag{A.H.}$$

$$= \varepsilon ,$$

where the first inequality holds from the given bounds on $l$ and $\rho$:

$$l - \rho = (1 - \eta)l + \frac{\eta l}{2} + \frac{\eta l}{2} - \rho$$

$$\geq (1 - \eta)l + \frac{\eta l}{2} \tag{since $\rho \leq \frac{\eta l}{2}$}$$

$$\geq (1 - \eta)l + \sqrt{2 \log(1/\varepsilon)l} . \tag{since $l \geq \frac{8}{\eta^2} \log(\frac{1}{\varepsilon})$}$$

This completes the proof of Claim 2.

We now combine Claims 1 and 2 to prove the theorem. Define $l_0 := \max(\frac{2}{\eta} \log n, \frac{8}{\eta^2} \log(\frac{1}{\varepsilon}))$. Define $m := \left\lceil \frac{\log n - \log \delta}{\eta^{l_0+1}} \right\rceil$. Note that $m$ is polynomial in $n, \delta, \varepsilon$.

Let $h$ denote the output of algorithm $\mathcal{A}$ given a sample $S \sim D^m$. We consider two cases. If $l \le l_0$ then, by Claim 1, $h = c$ (and hence the robust risk is 0) with probability at least $1 - \delta$. If $l_0 \le l$ then, since $\rho = \log n$, we have $l \ge \frac{8}{\eta^2} \log(\frac{1}{\varepsilon})$ and $\rho \le \frac{\eta l}{2}$ and so we can apply Claim 2. By Claim 2 we have

$$\mathsf{R}_\rho^E(h, c) \le \mathop{\mathbb{P}}_{x \sim D} (\exists z \in B_\rho(x) \cdot c(z) = 1) \le \varepsilon$$

$\square$

## Footnotes

[4]For a more detailed reasoning, we refer the reader to the proof of Theorem 9, where we bound the expected value $\mathbb{E}_{c,S}\left[\mathsf{R}_\rho^E(\mathcal{A}(S), c)\right]$ of the robust risk of a target chosen at uniformly random and the hypothesis outputted by a learning algorithm $\mathcal{A}$ on a sample $S$.