[Reviews · NeurIPS 2019]

Reviewer 1



1. Novelty and Significance: The paper mostly presents some impossibility results on robust binary classification under adversarial perturbation, which could be of independent interest for a mathematical perspective. However it has not been made clear how do these impossibility results have any impact from a practical point of view. I would have appreciated if authors have provided some algorithms which achieves robust learnability even for a slightly relaxed setting, e.g. at least a robust learning algorithm for monotone conjugate functions for Thm. 10's setting would have been more useful. 2. Writing: The writing/presentations of the paper requires a lot of polishing: a) I could not find the definition of "PAC-learnability" in the paper (used in Sec 5). b) The class of "Monotone conjugation function of length l" is nowhere defined, neither authors put any reference for this. c) What is meant by "non-trivial" function class? No definitions provided. d) What is the point of introducing Def. 2 as all the results are stated only for the risk measure defined in Def. 1 only. e) By "distribution free setting" -- does that imply a classification problem setting where instances do not come from any underlying distribution? f)There are lot of (minor) typos throughout the paper: Line 135 --> "fhas". Line 200 --> unexpected occurrence of ",", Line 239 "a universal" repeated, Line 310: given --> gave etc. etc. 3. Seemingly incomplete results/ confusing theorem statements: a) Its very confusing as authors claim in the Introduction that: "On the other hand, a more powerful learning algorithm that has access to membership queries can exactly learn monotone conjunctions and as a result can also robustly learn with respect to exact in the ball loss." --- I could not find details of these results anywhere in the paper later. b) Thm. 10 also seems to be confusing on the distributional assumption D -- does these include any distribution defined on the extreme points of the boolean hypercube? c) Statement of Prop. 12: I am surprised the sample complexity does not depend on the properties of the underlying distribution and k --- I expect sample complexity to go up with increasing k. Should it be |C'_{k,n}| instead of |C_{n}|? d) Another typo in Proposition 12: D' --> D'_{(k,n)}? e) Its also slightly disappointing to see that the sample complexity results holds only for finite size concept classes, precisely since authors only have assumed the instance set to be extreme points of the boolean hypercube. I would rather have appreciated the result lot more if proved for a more general concept class where sample complexity depends on some complexity measure (e.g. VC dimension) of C_n, etc. 4. Experiments: The paper does not provide any empirical studies, but I understand that this is rather out of the scope of the current results since all the claims are mostly made on non-existence of any robust-learning algorithms, but atleast authors could have proposed some algorithms for the monotone conjugate functions (i.e. for Thm. 10) with learnability guarantee to show the tightness of their analysis and simulated it.

Reviewer 2



* Line 47: "conceptual conceptual" * Line 67: redundant "examples" * Discussion of [5] (Bubeck et al.): I find the use of the word "computable" very confusing here. I couldn't understand why they would need to solve the Halting Problem... * Line 224: I can't find Equation (8) (also, please capitalize E in Equation)

Reviewer 3



This work by considering different efficiceny aspects of robust learning is new and brings important contributions to an important field of research. The positionning to recent related works is very clear stressing the limits of the state-of-the-art when integrating computational efficiency and sample complexity in the metrics of designing robust learning models. The article is theoretically sound and well written. I think this article is important in the field of robust learning by focusing on the efficiency of solutions and on what problems are then achievable when efficiency has to be considered. Their contributions open new directions to see whether this result can be generalised to other classes of functions than the simple monotone conjunctions addressed here and how state-of-the-art recent results hold when adding constraints on sample complexity and computational efficiency.

[Author Response · NeurIPS 2019]

We will fix all minor comments and typos without explicitly addressing them in the rebuttal.

**Response to Reviewer 1**:

*Practical Impact*: Our primary aim in this work is indeed theoretical. There has been substantial interest in the
theoretical understanding of adversarial robustness recently. Our work highlights the deficiencies in some of these
theoretical formulations (see also response to Reviewer 2 below), which we hope will lead to better theoretical models,
which in turn may lead to practical advances. Regarding an algorithm for monotone conjunctions in Theorem 10's
setting, the standard PAC learning algorithm for conjunctions suffices. An outline of this already appears in the
Appendix, but we will add a reference to it in the main paper.

*PAC Terminology*: We have assumed that readers will be familiar with standard terminology from PAC learning. Given
that many NeurIPS attendees may be unfamiliar with this terminology, we will add an appendix giving definitions that
we require and point readers to standard texts for further details.

*Non-trivial Class*: The definition of non-trivial class appears just before the statement of Theorem 5 (in lines 182-183).

*Undefined Algorithm*: The algorithm for *exact learning* monotone conjunctions using membership queries would be
considered folklore in the computational learning theory world; the key idea is that starting from the instance where all
bits are 1 (which is always a positive example), we can test whether each variable is in the target conjunction by setting
the corresponding bit to 0 and requesting the label. We will add this in the aforementioned new appendix.

*Finite Concept Classes*: Since (Thm. 11/Prop. 12) are primarily concerned with showing hardness of robust learning,
we don't think finite concept classes is a restriction. Please also see lines 70-90 for discussion regarding concept classes
defined over $\mathbb{R}^n$.

*Experiments*: We do not believe that *artificial* experiments will add to the value of the paper; that's not the main point
of the submission.

*Comparison to Prior Work/Contributions*: We will expand on the section in the paper, but we also refer to the review by
R3, which we believe very clearly summarizes our contributions.

**Response to Reviewer 2**: *Right Model*: We obviously disagree with the reviewer about this being a bad paper, but to a
great extent do agree with the reviewer about these being *unsuitable models* or *inadequate definitions* for adversarial
robustness. The point is that we *weren't* the inventors of these definitions (cf. [4, 5, 7, 8] for theory papers and others
more applied papers [A, B, C]). Our aim was precisely to show that once these definitions are *accepted*, even the most
elementary classes prove to be hard to robustly learn—and that proving computational hardness is much easier and
straightforward compared to the proofs that appeared in prior work.

Having criticized the definitions, we should acknowledge the contributions of prior work. Indeed, our initial aim was to
show positive results for at least some non-trivial classes under these definitions. It is clear that these definitions are in
many ways *natural* and *reasonable*, but when one looks at them under the lens of computational learning theory their
inadequacies surface immediately. We hope our work will highlight these issues and lead to future work (including
hopefully by us) that comes up with definitions that still (somewhat) retain the *simplicity* and *naturalness* of the current
definitions, while allowing one to separate non-trivial classes that are easy to robustly learn from those that are not!

*Computability/Halting Problem*: There is no connection to the halting problem, which is only one of the reasons why
uncomputability arises; there can be several others. The difficulty in this case is *enumeration* over (uncountably) infinite
sets. How would one compute the function, $\mathbf{1}(\exists y \in B_\rho(x).h(y) \neq c(y))$ in *finite time* even if one had black-box access
to evaluate $h$ and $c$ (the latter is not possible without membership queries)? Even under the assumption that the Turing
machine has the power to perform arithmetic over reals in unit time, the existential quantifier makes evaluating the
robust loss impossible! Even if the instance space is $\mathbb{Q}^n$, the decision problem for detecting an adversarial example
would be *recursively enumerable*, but not *recursive*. This problem disappears for finite instance spaces, but even there it
is not obvious how to evaluate this loss *without* membership queries. This is why one gets the separation for monotone
conjunctions depending on whether or not the learning algorithm has access to membership queries. In the case of
infinite instance spaces, we can't see a way to avoid the enumeration question without a strong inductive bias on $h$ and
$c$; in that case, properties of these functions, e.g. Lipschitzness, could be used to compute the loss in finite time.

**Response to Reviewer 3**:

We thank the reviewer for the comments and will obviously fix the typos they observed.

A. Cisse et al. Parseval Networks. ICML 2017.

B. Madry et al. Towards deep learning models that are resistant to adversarial attacks. ICLR 2018

C. Tramer et al. Ensemble Adversarial Training. ICLR 2018

[Meta-Review · NeurIPS 2019]

The present paper is about robust learnability, and important problem for our ML community. The authors provide both theoretical and methodological contributions to address sample complexity and computational efficiency in the robust learning framework. There are many results of importance in this paper, namely a nice characterization of what is the “strength” of an adversary, but the most interesting result is a negative one. It is a strong impossibility result saying that the class of monotone conjunctions is not efficiently robustly learnable when adversary can flip \omega(log(n)) bits. This paper is quite well written, although it will be greatly improved if the authors manage to add the content of their rebuttal in the camera ready version. Because of these days importance of the subject, and because of the quality of the results, I am recommending acceptation with a short talk.